# Breast Cancer Brain Metastasis—Overview of Disease State, Treatment Options and Future Perspectives

**DOI:** 10.3390/cancers13051078

**Published:** 2021-03-03

**Authors:** Chikashi Watase, Sho Shiino, Tatsunori Shimoi, Emi Noguchi, Tomoya Kaneda, Yusuke Yamamoto, Kan Yonemori, Shin Takayama, Akihiko Suto

**Affiliations:** 1Department of Breast Surgery, National Cancer Center Hospital, Tokyo 104-0045, Japan; cwatase@ncc.go.jp (C.W.); sshiino@ncc.go.jp (S.S.); stakayam@ncc.go.jp (S.T.); 2Department of Breast and Medical Oncology, National Cancer Center Hospital, Tokyo 104-0045, Japan; tshimoi@ncc.go.jp (T.S.); enoguchi@ncc.go.jp (E.N.); kyonemor@ncc.go.jp (K.Y.); 3Department of Radiation Oncology, National Cancer Center Hospital, Tokyo 104-0045, Japan; tomkaned@ncc.go.jp; 4Division of Cellular Signaling, National Cancer Center Research Institute, Tokyo 104-0045, Japan; yuyamamo@ncc.go.jp

**Keywords:** central nervous system, blood–brain barrier, neurosurgery, stereotactic radiosurgery (SRS), whole brain radiation therapy (WBRT), molecular-targeted therapy, tyrosine kinase inhibitor, CDK4/6 inhibitor, immune checkpoint inhibitor, BRCA gene mutation, review

## Abstract

**Simple Summary:**

In this review, we present the latest information on the pathophysiology, diagnosis, and local and systemic treatment of brain metastases from breast cancer, with a focus on recent publications. Improving the local treatment and subtype-specific systemic therapies through advancements in basic and translational research will contribute to better clinical outcomes for patients with breast cancer brain metastasis.

**Abstract:**

Breast cancer is the second most common origin of brain metastasis after lung cancer. Brain metastasis in breast cancer is commonly found in patients with advanced course disease and has a poor prognosis because the blood–brain barrier is thought to be a major obstacle to the delivery of many drugs in the central nervous system. Therefore, local treatments including surgery, stereotactic radiation therapy, and whole-brain radiation therapy are currently considered the gold standard treatments. Meanwhile, new targeted therapies based on subtype have recently been developed. Some drugs can exceed the blood–brain barrier and enter the central nervous system. New technology for early detection and personalized medicine for metastasis are warranted. In this review, we summarize the historical overview of treatment with a focus on local treatment, the latest drug treatment strategies, and future perspectives using novel therapeutic agents for breast cancer patients with brain metastasis, including ongoing clinical trials.

## 1. Introduction

Breast cancer brain metastasis (BCBM) is one of the most common forms of breast cancer metastasis [1,2]. Several articles have investigated the incident rate of BCBM so far. Barnholtz-Sloan et al. reported that the incidence rate of BCBM was 5.1% among patients with breast cancer. Furthermore, 14.2% developed BCBM during the clinical course of disease among patients with any distant metastases [1]. Several other articles have also reported the incidence rates of BCBM [3,4]. The incidence rates for BCBM depend on the cancer subtype. In particular, HER2-type (hormone receptor (HR)-negative/ human epidermal growth factor receptor 2 (HER2)-positive) and triple-negative breast cancer (TNBC: HR-negative/HER2-negative) subtypes are more likely to metastasize to the brain. Martin et al. reviewed more than 230,000 breast cancer patients and reported the incidence of brain metastases by cancer subtype as follows: 0.22%, 0.61%, 1.09%, and 0.68% in HR-positive/HER2-negative patients, HR-positive/HER2-positive patients, HER2-type patients, and TNBC patients, respectively. The respective incidence rates in patients with systemic metastasis at the time of initial diagnosis were 5.46%, 7.98%, 11.45%, and 11.37% [4]. Several other retrospective analyses have been conducted and reported BCBM frequencies of 14–38% in HER2-type and TNBC patients and less than 10% in patients with the luminal-type [5,6,7,8,9,10]. Especially, HER2-type increased the risk of BCBM when compared with the other cancer subtypes in the multivariate analysis [5,11].

Some studies have reported various risk factors including cancer subtype associated with the incidence of BCBM. Recently, Koniali et al. performed a systematic review to identify the risk factors of BCBM [12]. They found that younger age, estrogen receptor (ER)-negative status, HER2-positive status, higher tumor stage, higher histologic grade (HG), large tumor size, and high Ki67 labeling index were independent risk factors for BCBM. In other reports, the following various risk factors have been reported: ER-negative status [13,14], younger age [5,14], HER2-positive status [5,11,14,15], basal-like type [11], triple-negative non-basal type [11], axillary LN metastasis [15], higher HG [14,16], higher stage [17], number of extracranial metastatic sites [14], and short disease-free survival (DFS) [14].

The prognosis for BCBM depends on the breast cancer subtype. The median overall survival (OS) was 7.1 months for HR-positive/HER2-negative, 18.9 months for HR-positive/HER2-positive, 13.1 months for HER2-type, and 4.4 months for TNBC [18]. Niikura et al. reported that the median OS was 8.7 months; when divided by subtype, luminal-type had an OS of 9.3 months, luminal-HER2 type had an OS of 16.5 months, HER2 type had an OS of 11.5 months, and TNBC type had an OS of only 4.9 months [19]. Thus, the treatment strategy according to breast cancer subtype should be considered to improve survival. Some prognostic tools have already been established to evaluate prognosis for patients with BCBM. The Diagnosis-Specific Graded Prognostic Assessment (GPA) prognostic model, including Karnofsky Performance Status (KPS) plus age, can distinguish patients with a two-year median survival versus those with a median survival of 3.4 months [20,21,22]. However, modified breast-GPA are more accurate predictive indexes for BCBM [22].

The prognosis after brain metastasis remains poor as discussed above, because the blood–brain barrier (BBB) removes drug substances, including chemotherapeutic agents, targeted agents, and toxins, from the brain [23,24,25]. The BBB is constructed of specialized blood vessel structures and is comprised of endothelial cells, astrocytes, pericytes, and neurovascular units [26]. The endothelial cells include several transporters (P-glycoprotein, multidrug-resistance proteins, etc.) and act as efflux pumps. Consequently, Pardridge et al. have mentioned that for drug molecules to cross the BBB, they must be under 400–500 Da in size and have a high lipid solubility [27]. The BBB is impaired by brain metastasis (BM), which leads to the creation of the blood–tumor barrier (BTB) [28,29]. Lockman et al. and Gril et al. demonstrated that the BTB limits the uptake of chemotherapeutic drugs [30,31] whereas most chemotherapeutic agents could enter central nervous system (CNS) lesions via the intact BBB. Meanwhile, trastuzumab has been shown to penetrate experimental BCBMs [32,33]. ^89^Zr trastuzumab [34], ^14^C-paclitaxel, ^14^C-doxorubicin [30], and ^14^C-lapatinib [35] had been demonstrated to concentrate in BCBM. Morikawa et al. found that capecitabine and lapatinib in BCBMs penetrate the BTB in actual patients using a liquid chromatography tandem mass spectrometry method [36]. These results suggest BBB and BTB permeability enable some drug agents to access BCBM, therefore those drug agents are desired for BCBM therapy.

In this review, we present the latest information on the many clinical challenges in the treatment strategies for BCBM. Local therapies, including surgery and radiotherapy, have been improved to be less invasive and to allow the patient to retain more cognitive function. Recently, subtype-specific systemic therapies that have been developed for breast cancer and genomic sequencing data for BCBM can contribute to genomically guided treatment for target mutated genes. Furthermore, the association of microRNA (miRNA) or circulating tumor DNA (ctDNA) with BCBM has been investigated, and these molecules have the potential to play roles in future diagnosis and treatment.

## 2. Local Treatment for Patients with BCBM

Neurosurgical resection, stereotactic radiosurgery (SRS), and whole brain radiation therapy (WBRT) are the current standard local therapy treatments for BCBM. The local treatment strategies for BCBM depend on the patients’ performance status and number/size/location of the brain tumor [37,38]. Table 1 (major clinical trials) and Appendix A
Appendix A (other clinical trials) show main clinical trials in radiation therapy.

### 2.1. Patients with Single or Two to Four BMs

#### 2.1.1. Surgery Plus WBRT

Surgery is performed to retrieve metastatic tissue for pathologic confirmation, improvement of mass effect and edema [50]. However, surgery alone is considered inadequate for local control compared with surgery plus radiotherapy [40]. For operable single BCBM, three randomized trials have demonstrated that surgery plus WBRT contribute to improved clinical outcomes compared with WBRT alone [39,51,52]. Churilla et al. reported that SRS shows similar local control compared with surgery for one to two brain metastases under 4 cm in size (hazard ratio, 1.15; 95% CI, 0.72–1.83) [53].

#### 2.1.2. Postoperative SRS vs. WBRT

Some randomized trials have compared postoperative SRS to the surgical cavity and WBRT. Kępka et al. compared SRS of the tumor bed with WBRT after resection of a single BM [54]. However, this trial was underpowered for the detection of noninferiority of SRS of the tumor bed compared with surgery with WBRT.

In the NCCTG N107C/CEC·3 cooperative group study (NCT01372774), there was no difference in the median OS between postoperative SRS and WBRT (hazard ratio, 1.07; 95% CI, 0.76–1.50; *p* = 0.70), but postoperative SRS resulted in superior cognitive function for patients with one resected BM (up to three unresected metastases were allowed) and a resection cavity less than 5.0 cm (hazard ratio, 0.47; 95% CI, 0.35–0.63; *p* < 0.0001) [41]. Recently, Kayama et al. evaluated whether salvage SRS alone within 21 days of surgery is as effective as postoperative WBRT on the OS of patients with one to four BMs in a noninferiority, randomized controlled trial (JCOG0504) [42]. They concluded that salvage SRS, which is noninferior to postoperative WBRT (hazard ratio, 1.05; 90% CI, 0.83–1.33; one-sided *p* for noninferiority = 0.027), can be a standard therapy for patients with one to four BMs.

#### 2.1.3. WBRT Alone versus WBRT Plus SRS

Some studies have investigated the clinical differences between WBRT alone and WBRT plus SRS for patients with one to three BMs. The RTOG9508 trial compared the survival between WBRT alone and WBRT followed by SRS in patients with one to three BMs [43]. They demonstrated that the mean survival time did not differ between those two groups, but WBRT plus SRS resulted in better survival for patients with a single unresectable BM (median survival time, 6.5 (WBRT plus SRS) versus 4.9 months (WBRT); *p* = 0.0393). Kondziolka et al. reported that there was no difference in OS between those groups (*p* = 0.22), but the median time to local failure was longer in the WBRT plus SRS group compared with the SRS group (36 (WBRT plus SRS) versus 6 (SRS) months; *p* = 0.0005) [55]. Meanwhile, Tsao et al. performed a meta-analysis evaluating WBRT versus WBRT plus SRS for those two studies. There was no difference in OS between those two groups (*p* = 0.24), but local control favors WBRT plus SRS (hazard ratio 2.88; 95% CI, 1.63–5.08; *p* = 0.0003) [56].

#### 2.1.4. SRS Only versus WBRT Plus SRS

Several previous clinical trials have investigated the difference between SRS only and WBRT plus SRS [44,45,46]. Aoyama et al. reported that there was significantly decreased brain tumor recurrence in the WBRT plus SRS group; however, WBRT plus SRS did not improve the survival for patients with one to four BMs (*p* = 0.42). Therefore, SRS alone could be a treatment option, provided that the frequent monitoring of brain tumor status is conducted [46]. The EORTC22952–26001 study investigated the clinical utility of adjuvant WBRT after either surgery or SRS for patients with one to three BMs. In this study, the OS between WBRT and observation did not differ significantly (*p* = 0.89), but adjuvant WBRT reduced the relapse rate both at initial sites and at new sites (from 31% to 19%; *p* = 0.040 (initial sites); from 48% to 33%; *p* = 0.023 (new sites)) [44]. Brown et al. also reported that the OS did not differ between SRS alone and SRS plus WBRT for patients with one to three BMs (hazard ratio, 1.02; 95% CI, 0.75–1.38; *p* = 0.92), but the time to intracranial failure for the SRS alone group was significantly shorter compared with the SRS plus WBRT group (hazard ratio, 3.6; 95% CI 2.2–5.9; *p* < 0.001) (NCT00377156) [45].

However, some studies have reported problematic findings with WBRT, including a worsened quality of life [57] and neuro dysfunctions [45,58]. Chang et al. investigated whether adding WBRT to SRS could be associated with potential neuro-cognitive dysfunction as a primary endpoint in a randomized controlled trial [58]. They concluded that the SRS plus WBRT group displayed significantly decreased learning and memory functions at four months compared with the SRS alone group (the mean posterior probabilities of decline were 52% (SRS plus WBRT) and 24% (SRS alone)). Brown et al. also found that the incidence of cognitive deterioration was lower in the SRS alone group at both 3 and 12 months compared with the SRS plus WBRT group (45.5% (SRS alone) versus 94.1% (SRS plus WBRT); *p* = 0.0017 at 3 months; 60% (SRS alone) versus 94.4% (SRS plus WBRT); *p* = 0.04 (12 months)) [45]. Therefore, they concluded that SRS alone could be considered a better treatment strategy for patients with one to three BMs compared with the combination of WBRT and SRS.

### 2.2. Patients with Five or More BMs

The differences between WBRT and SRS remain unclear for patients with five or more BCBMs. Therefore, a phase III randomized controlled trial (NCT01592968) compared SRS with WBRT for patients with 4–15 untreated non-melanoma BMs [48]. There was no difference in the median OS between the two groups (10.4 months in the SRS group and 8.4 months in the WBRT group, *p* = 0.45). Moreover, SRS reduced the risk of neuro-cognitive deterioration in patients with 4–15 BMs. Therefore, this study suggests that avoiding WBRT may be possible for patients with more than ten metastases. Yamamoto et al. reported that there was no difference in the median OS between patients with 2–4 tumors and 5–10 tumors (hazard ratio, 0.97; 95% CI, 0.81–1.18 (less than noninferiority margin); *p* = 0.78; *p* for noninferiority < 0.0001) in the JLGK0901, which evaluated patients who received SRS for BMs (10% of such patients had breast cancer) [47]. The incidence rate of one or more treatment-related adverse event did not differ between patients with 2–4 or 5–10 tumors (*p* = 0.89). Thus, SRS without WBRT could be an alternative strategy for patients with five to ten BMs. In a long-term follow-up study, the neuro-cognitive function did not differ between those two groups [59]. Meanwhile, the NCT03075072 study is ongoing to investigate quality of life between HA-WBRT (Hippocampal-avoidance WBRT) and SRS in patients with 5–20 BM tumors. The NCT04061408 study, a phase II trial in China, is currently investigating the control rate for 1–10 BM lesions in BC patients using fractionated stereotactic radiotherapy (FSRT).

### 2.3. Preoperative SRS vs. Postoperative SRS

In a multi-institutional retrospective analysis comparing preoperative SRS with postoperative SRS [60], there was no difference between the two groups in OS (*p* = 0.1), local recurrence (*p* = 0.24), and distant brain recurrence (*p* = 0.75) in the multivariable analysis; however, the incidences of symptomatic radiation necrosis (2 years: 16.6% (postoperative SRS) vs. 3.2% (preoperative SRS); *p* = 0.010) and leptomeningeal disease (2 years: 16.4% (postoperative SRS) vs. 4.9% (preoperative SRS); *p* = 0.010) were significantly lower in the pre-SRS groups. Additionally, some clinical trials for preoperative SRS are ongoing (NCT03741673: preoperative SRS vs. postoperative SRS; NCT03368625: preoperative SRS phase II trial).

### 2.4. Hippocampal-Avoidance WBRT (HA-WBRT) for BMs

The effectiveness of hippocampal-avoidance WBRT (HA-WBRT) in protecting against the neuro-cognitive toxicity of standard WBRT has been investigated recently [49,61]. RTOG0933, a phase II multi-institutional trial, demonstrated that HA-WBRT helps preserve neuro-cognitive function and quality of life compared with historical controls (*p* < 0.001) (NCT01227954) [49]. Additionally, Sun et al. reported that HA-WBRT is considered appropriate because BCBM has a low risk of metastases and recurrence at the hippocampal avoidance region [62]. Meanwhile, NRG Oncology CC001, a phase III trial (NCT02360215), enrolled adult patients with BMs into either HA-WBRT plus memantine or WBRT plus memantine groups. The study demonstrated a lower risk of neuro-cognitive failure after HA-WBRT plus memantine compared with WBRT plus memantine (adjusted hazard ratio, 0.74; 95% CI, 0.58–0.95; *p* = 0.02). Additionally, there were no differences in OS, intracranial progression-free survival (PFS), or toxicity [61]. Therefore, HA-WBRT could be a better option for preserving neuro-cognitive function. Memantine, an N-methyl-D-aspartate receptor antagonist, has been considered to reduce neuro-cognitive decline from WBRT because radiotherapy for cerebral substance could lead to cognitive deterioration as a result of the radiation-induced accelerated atherosclerosis and microangiopathy and infarction [63]. Brown et al. reported that memantine was well-tolerated and resulted in better neuro-cognitive function in patients receiving WBRT (*p* = 0.0041 at 16 weeks) [64].

## 3. Systemic Therapy for BCBM

Table 2 (major clinical trials) and Appendix A (other clinical trials) show the main clinical trials in systemic therapy according to subtype.

### 3.1. Systemic Therapy for HER2-Positive BCBM

#### 3.1.1. Trastuzumab for BCBM

Trastuzumab is a monoclonal antibody with a large molecular weight that makes it difficult to cross the BBB, and normal intravenous administration is not considered effective for BCBM. Trastuzumab has not been considered crossing the intact BBB. However, Lewis et al. showed the uptake of ^89^Zr-trastuzumab in HER2-positive brain tumors of mouse models [32]. Meanwhile, Tamura et al. showed that trastuzumab accumulates in BCBMs by using ^64^Cu-DOTA-trastuzumab with positron emission tomography (PET) imaging to visualize and quantify HER2-positive lesions in patients with HER2-positive breast cancer [88].

Park et al. investigated the role of trastuzumab for patients with HER2-positive BCBM. In this study, patients receiving trastuzumab for BCBM had a significantly longer time to death after BCBM [89]. A multicenter prospective study (registHER) revealed that trastuzumab treatment after the first CNS diagnosis significantly decreased the risk of death [7]. In a study using data from the HERA trial, there were no differences between the group given one year of adjuvant trastuzumab and the observational group in the frequency of CNS metastases in patients with HER2-positive breast cancer [90].

Intrathecal trastuzumab was administered in several studies in breast cancer patients with leptomeningeal disease [91]. In a case report, Bousquet et al. observed the efficacy of intrathecal trastuzumab injections for HER2-positive breast cancer with leptomeningeal metastasis [92]. The NCT02571530 study aims to evaluate the safety of super-selective intra-arterial cerebral infusions of trastuzumab. This method is a promising treatment for patients with leptomeningeal dissemination of HER2-positive breast cancer, but it will be some time before it is applied in clinical practice because the study population is small, enrollment in the ongoing clinical trials is poor, and the safety has not been sufficiently reported.

#### 3.1.2. Lapatinib for BCBM

Lapatinib, which is a small molecule dual tyrosine kinase inhibitor of the EGFR and HER2 tyrosine kinases, can cross the BBB because of its very low molecular weight (581 Da) [30,35]. Therefore, it may be effective for BM. Taskar et al. investigated the distribution of lapatinib in CNS lesions using ^14^C-lapatinib administration. They concluded that BTB permeability is crucial for the distribution of lapatinib [35]. In a phase II trial, Lin et al. demonstrated a CNS objective response (defined as complete response (CR) plus partial response (PR)) rate of 6% in the lapatinib monotherapy group [65]. The additional response rate was confirmed in the group receiving lapatinib and capecitabine.

The usefulness of lapatinib in combination with capecitabine has been reported in several trials. In a phase III trial (the EGF100151 trial), lapatinib plus capecitabine was compared with capecitabine alone in advanced breast cancer treated with chemotherapy and trastuzumab-containing regimens. The addition of lapatinib significantly prolonged the time to progression and was associated with fewer cases of CNS metastasis as the first progression [67,93]. A multicenter phase II study of lapatinib in patients with HER2-positive BCBM (the EGF105084 trial) showed the effect of lapatinib and capecitabine combination therapy for patients resistant to lapatinib monotherapy. The response rate was 20%, and volumetric reduction was observed in the CNS lesions of the patients treated with the combination of lapatinib and capecitabine [65]. The single-arm phase II LANDSCAPE trial investigated the efficacy of the combination therapy of lapatinib and capecitabine for patients with an initial recurrence of BMs not previously treated with WBRT (NCT00967031) [66]. In that study, the CNS response rate was 65.9% (29 of the 44 assessable patients). The phase III randomized CEREBEL study was designed to investigate the incidence of CNS as the first progression in patients treated with lapatinib plus capecitabine (NCT00820222) [94]. In that study, lapatinib plus capecitabine was compared with trastuzumab plus capecitabine in patients with HER2-positive metastatic breast cancer (MBC). Trastuzumab plus capecitabine had a longer PFS (hazard ratio, 1.30; 95% CI, 1.04–1.64; significant) and OS (HR, 1.34; 95% CI, 0.95–1.64; not significant). However, the CEREBEL study did not meet its primary endpoint because of insufficient CNS events. Kaplan et al. reported the effects of lapatinib plus capecitabine in patients with HER2-positive BCBM. The clinical benefit rate (PR or stable disease (SD)) was 68.4%, and this regimen was an independent predictor for better survival in the multivariate analysis [95].

The usefulness of lapatinib in combination with trastuzumab has also been investigated. In a retrospective analysis of HER2-positive metastatic or recurrent breast cancer with brain metastasis, patients treated with both trastuzumab and lapatinib after developing metastasis had a significantly longer survival than patients treated with trastuzumab alone, lapatinib alone, or no HER2-targeting agent (*p* < 0.001) [96]. Conversely, in the results of the NCIC CTG MA.31 phase II trial investigating taxane plus lapatinib or trastuzumab as the first-line treatment for HER2-positive breast cancer, the lapatinib plus taxane group had a significantly shorter PFS (hazard ratio, 1.48; 95% CI, 1.20–1.83; *p* < 0.001) and more toxicity than the trastuzumab plus taxane group (NCT00667251) [97].

In an in vivo experiment, Sambade et al. reported that lapatinib leads to radiosensitivity [98]. In that study, the inhibition of ERK1/2 and AKT was correlated with lapatinib-mediated radio sensitization. The combination of lapatinib and WBRT has also been evaluated in some clinical studies [68]. In that study, lapatinib with WBRT had a higher objective response rate (79%) compared with historical controls of WBRT alone. Therefore, the RTOG group performed a phase II trial to investigate the effect of lapatinib with radiation therapy (WBRT or SRS) in patients with HER2-positive BCBM (NCT01622868). Another phase II trial is also being performed to evaluate the response rate of BMs to WBRT and lapatinib (NCT01218529).

#### 3.1.3. Pertuzumab for BCBM

Pertuzumab is a humanized monoclonal antibody that inhibits the dimerization of HER2 with other HER receptors. Pertuzumab is also considered a systemic treatment for CNS metastases. The CLEOPATRA trial is a phase III study to compare pertuzumab, trastuzumab, and docetaxel with placebo, trastuzumab, and docetaxel (NCT00567190) [99]. In the exploratory analyses for this trial, Swain et al. have investigated the incidence and time to development of CNS metastases [69]. The incidence of CNS disease was delayed in the groups with the addition of pertuzumab compared with placebo, trastuzumab, and docetaxel (hazard ratio, 0.58; 95% CI, 0.39–0.85; *p* = 0.0049). However, the incidence of BCBM was similar between the pertuzumab and placebo groups (13.7% (pertuzumab) vs. 12.6% (placebo)).

The PATRICIA study (phase II study: NCT02536339) is a currently ongoing study examining the safety and efficacy of pertuzumab in combination with high-dose trastuzumab in patients with HER2-positive CNS metastases who have CNS progression following radiation therapy. An interim analysis of this study was recently reported [100].

In the NCT02598427 phase I clinical trial, the evaluation of intrathecal pertuzumab and trastuzumab was planned; however, the trial was terminated because of inadequate enrollment.

#### 3.1.4. Trastuzumab Emtansine (T-DM1) for BCBM

T-DM1 is an antibody drug conjugate composed of the monoclonal antibody trastuzumab linked to the cytotoxic agent DM1 (maytansine derivative). Some articles have reported case series that administered T-DM1 to patients with HER2-positive BCBM [101,102] and reported that T-DM1 was a well-tolerated treatment strategy for patients with HER2-positive BCBM. In the exploratory analysis of the phase III EMILIA trial, which compared T-DM1 with capecitabine and lapatinib, the rate of CNS progression was similar between T-DM1 and capecitabine plus lapatinib. In patients with treated, asymptomatic CNS metastasis at baseline, OS was improved in the T-DM1 group compared with lapatinib plus capecitabine (hazard ratio, 0.38; *p* = 0.008; median 26.8 versus 12.9 months) [70].

The KAMILLA trial, a single-arm phase IIIb study of T-DM1 in patients with HER2-positive locally advanced/metastatic breast cancer with prior HER2-targeted therapy and chemotherapy, showed the efficacy and safety of T-DM1 in HER2-positive BCBM. This trial showed an overall response rate of 21.4%, a PFS of 5.5 months, and an OS of 18.9 months in patients with BCBM (NCT01702571) [71].

The exploratory analysis of the KATHERINE trial, a phase III trial, compared adjuvant T-DM1 and trastuzumab for patients who had residual invasive disease after the completion of neoadjuvant therapy; the incidence of CNS recurrence as the first invasive-disease event at the follow-up of 3 years was 4.3% in the trastuzumab arm and 5.9 % in the T-DM1 arm, respectively [103].

#### 3.1.5. Neratinib for BCBM

Neratinib is an oral small molecule irreversible inhibitor of tyrosine kinase activity including that of EGFR, HER1, HER2, and HER4 [104,105]. This agent demonstrated clinical utility as both a single agent [105] and in combination with paclitaxel [106].

In the randomized phase III trial, ExteNET, demonstrated that 1 year of adjuvant neratinib after chemotherapy plus trastuzumab contributes to significant better PFS in operable breast cancer (hazard ratio, 0.73; 95% CI 0.57–0.92, *p* = 0.0083) [107]. The cumulative incidence of CNS recurrences was fewer in the neratinib group (0.7% with neratinib, 2.1% with placebo, respectively) in the HR-positive patients who initiated the study treatment within 1 year of prior trastuzumab [108].

The NEfERT-T phase II randomized trial investigated the efficacy and safety of neratinib plus paclitaxel compared with trastuzumab–paclitaxel in HER2-positive MBC (NCT00915018) [72]. In this study, the incidence of CNS metastases was lower (relative risk, 0.48; 95% CI, 0.29–0.79; *p* = 0.002) and the time to CNS metastases was delayed (hazard ratio, 0.45; 95% CI, 0.26–0.78; *p* = 0.004) in the group receiving neratinib plus paclitaxel.

In the TBCRC 022 trial, the combination of neratinib plus capecitabine showed a CNS response rate of 49% in the lapatinib-naïve group against 33% for the lapatinib-treated group (NCT01494662) [73]. The NALA trial (NCT01808573) [74], which explored the effect of adding neratinib or lapatinib to capecitabine (including 16.6% of patients with stable BM), demonstrated better survival in the neratinib group than in the lapatinib group. The hazard ratio for PFS was 0.76 (95% CI, 0.63 to 0.93; stratified log rank *p* = 0.0059), and the hazard ratio for OS was 0.88 (95% CI, 0.72 to 1.07; *p* = 0.2098). In the NALA trial, the overall cumulative incidence of intervention for CNS disease was 22.8% (95% CI, 15.5–30.9%) for neratinib versus 29.2% (95% CI, 22.5–36.1%) for lapatinib (*p* = 0.043), respectively.

#### 3.1.6. Tucatinib for BCBM

Tucatinib is a highly specific HER2-targeted tyrosine kinase inhibitor with minimal inhibition of the epidermal growth factor receptor [76]. Tucatinib combined with trastuzumab and capecitabine showed increased CNS response rates and better PFS rates. In the HER2CLIMB phase III trial (NCT02614794), which explored the impact of tucatinib combined with trastuzumab and capecitabine on intracranial efficacy and survival in patients with HER2-positive MBC with BMs, tucatinib showed a statistically significant improvement in PFS (hazard ratio for disease progression or death, 0.54; 95% confidence interval (CI), 0.42 to 0.71; *p* < 0.001) [76]. The addition of tucatinib to trastuzumab and capecitabine doubled the intracranial objective response rate (40.6% versus 22.8%; *p* < 0.001), and reduced the risk of intracranial progression or death (hazard ratio, 0.48; 95% CI, 0.34–0.69; *p* < 0.001). Meanwhile, the NCT03975647 study (HER2CLIMB-02) is evaluating the efficacy and safety of tucatinib in combination with T-DM1 in patients with HER2-positive MBC.

#### 3.1.7. Trastuzumab Deruxtecan for BCBM

Trastuzumab deruxtecan is an antibody–drug conjugate containing trastuzumab and exatecan derivative (topoisomerase I inhibitor). A phase II trial DESTINY-Breast01 investigated the efficacy and safety of trastuzumab deruxtecan for HER2-positive MBC previously treated with T-DM1 (NCT03248492). The objective response rate was 60.9% and the median PFS was 16.4 months [109].

### 3.2. Systemic Therapy for Luminal-Type BCBM

#### 3.2.1. Endocrine Therapy

CNS metastases occur less frequently for luminal-types compared with other subtypes [1,21]. Only case reports of successful cases have been reported regarding the effects of endocrine therapy (tamoxifen, megesterol acetate, letrozole, fulvestrant) on BM for ER-positive and HER2-negative breast cancer, and efficacy has not been confirmed in clinical trials [110].

#### 3.2.2. PI3K Inhibitor for BCBM

Approximately 40% of HR-positive, HER2-negative breast cancer displays *PIK3CA* mutations. Phosphatidylinositol 3-kinase (PI3K) inhibition has shown antitumor activity [111]. Le Rhun et al. reported that the *PI3KR1-rs706716* gene may be associated with CNS metastasis (NCT00959556) [112]. Chen et al. reported that PI3K inhibition with buparlisib and alpelisib sensitized ER-positive breast cancer cell lines to tamoxifen [113].

The phase III SOLAR-1 clinical trial of alpelisib plus fulvestrant in HR-positive/HER2-negative MBC showed better PFS in the alpelisib–fulvestrant group in the cohort of patients with *PIK3CA*-mutated cancer (hazard ratio, 0.65; 95% CI, 0.50–0.85; *p* < 0.001) (NCT02437318) [79]. Even in patients with BMs, four cases of reduced size or SD have been reported with the use of alpelisib in combination with hormone therapy [114].

The phase III clinical trial SANDPIPER (NCT02340221) comparing taselisib or placebo for ER-positive/*PIK3CA*-mutant MBC showed significantly longer investigator-assessed median PFS with taselisib (7.4  versus 5.4 months; hazard ratio, 0.70; *p* = 0.0037) (NCT02340221) [78]

Buparlisib, a pan-PI3K inhibitor, displayed a better PFS (hazard ratio, 0.67; 95% CI, 0.53–0.84; one-sided *p* = 0.0003); however, increased toxicity has also been demonstrated in the subsequent phase III BELLE-3 trial (NCT01633060) of buparlisib plus fulvestrant for MBC patients [80].

Recent studies evaluating metastatic organs by gene transfer to cell lines have shown that *PIK3CA* is an important gene mutation associated with BM [115]. Thus, the development of therapy for *PIK3CA* mutations may directly lead to the development of therapy for BM.

#### 3.2.3. CDK4/6 Inhibitors for BCBM

CDK4/6 inhibitors block the aberrantly accelerated cell cycle transition from the G1 to S phase. This transition is regulated by a complex consisting of CDK4/6 and cyclin D, which phosphorylates Rb and introduces E2F release; therefore, CDK4/6 inhibitors suppress cell cycle dysregulation.

Many articles have evaluated the CDK4/6 inhibitors palbociclib, ribociclib, and abemaciclib in advanced breast cancer [116,117,118]. An open-label phase II trial of abemaciclib has been reported (NCT02308020) [119]. This trial divided patients into four groups: cohorts A (HR-positive, HER-negative MBC), B (HR-positive, HER2-positive MBC), C (HR-positive MBC with leptomeningeal metastases), and D (BM treated with surgical resection). In cohort A, the objective response rate of intracranial lesions was 5.2%, and the intracranial clinical benefit rate was 24% with a median PFS of 4.9 months with some long responses. In cohort D, abemaciclib achieved therapeutic concentrations in BM tissue; therefore, abemaciclib and its metabolites can cross the BBB.

A single-arm phase II trial (NCT02774681) has evaluated palbociclib plus trastuzumab for patients with HR-positive, HER2-positive BCBM [120]. A phase II trial to evaluate the efficacy and safety of palbociclib in recurrent BCBM is ongoing (NCT02896335). The NCT04334330 study has evaluated the efficacy of palbociclib, trastuzumab, and lapatinib with fulvestrant in patients with ER-positive/HER2-positive BCBMs. Additionally, a phase I trial has been conducted to study the side effects of SRS with abemaciclib, ribociclib, or palbociclib in patients with HR-positive BCBM (NCT04585724).

### 3.3. Systemic Therapy for Triple-Negative BCBM

#### 3.3.1. Immune Checkpoint Inhibitors for BCBM

The programmed cell death protein 1 receptor (PD-1), its ligand (PD-L1), and cytotoxic T lymphocyte antigen 4 (CTLA-4) are inhibitory immune checkpoint receptors expressed on the surface of T cells, NK (natural killer) cells, B cells, macrophages, and dendritic cells. Several of these inhibitor agents have been investigated for targeting in breast cancer (PD-1 inhibitors: pembrolizumab and nivolumab; PD-L1 inhibitors: atezolizumab, avelumab, and durvalumab; CTLA-4 inhibitor: ipilimumab and tremelimumab). Duchnowska et al. showed that PD-L1 expression on tumor-infiltrating lymphocytes (TILs) was an independent favorable factor [121]. They suggest that this factor could be a potential therapeutic target of immune checkpoint inhibitors in BCBM. PD-L1 expression and tumor mutational burden are more frequently found in HER2-positive subtypes compared with luminal subtypes [122,123]. Narloch et al. compared the TIL rate between matched primary breast cancer (PBC) and BCBM. The percentage of TILs was lower in metastasis compared with PBC, especially TNBC showed the most decrease (5% in BCBM, 20% in PBC; *p* = 0.022) [124]. Li et al. reported a higher prevalence of PD-L1 expression on immune cells in the brain compared to the liver and bone (50.0% in the brain, 26.9% in the liver, and 25.0% in bone) [125], however, the number of patients with BMs is limited to four and it is not a definitive.

In several trials, PD-L1 inhibitors have been considered as targeted treatments for advanced breast cancer. The KEYNOTE-012 trial (NCT01848834) demonstrated that pembrolizumab had acceptable safety and activity for advanced TNBC [81]. The IMpassion130 trial (NCT02425891) aimed to evaluate the efficacy and safety of atezolizumab plus nab-paclitaxel compared with nab-paclitaxel for patients with locally advanced or metastatic TNBC [82]. The JAVELIN Solid Tumor study, a phase I trial, demonstrated clinical activity of avelumab in MBC [126]. A new anti-PD-L1 antibody (SHR-1316) in combination with cisplatin/carboplatin and bevacizumab was investigated in patients with HER2-type and triple-negative BCBM (NCT04303988).

Moreover, immune checkpoint inhibitors have been investigated in combination with radiotherapy. An atezolizumab phase II trial evaluated the combination of SRS with atezolizumab for TNBC with BCBM (NCT03483012). A phase I/II trial to investigate the efficacy and safety of pembrolizumab and SRS of patients with BCBM is ongoing (NCT03449238). In a phase I (NCT03807765) study, SRS after nivolumab was evaluated in patients with BCBM. Furthermore, a randomized phase II trial to explore the efficacy of nivolumab and ipilimumab for previously untreated, surgically-resectable, solid tumor BMs is planned (NCT04434560).

#### 3.3.2. Bevacizumab for BCBM

Bevacizumab is a monoclonal antibody against vascular endothelial growth factor. In a phase II trial evaluating the efficacy of carboplatin and bevacizumab for patients with BCBM (29 HER2-positive and 9 HER2-negative), the objective response rate was 63% (95% CI, 46–78%), the median PFS was 5.62 months, and the median OS was 14.10 months [83]. Furthermore, a phase II trial (NCT01281696) is investigating the efficacy of bevacizumab, cisplatin, and etoposide in patients with CNS metastasis.

### 3.4. New Targeted Agents for BCBM

#### PARP Inhibitors for BCBM

PARP inhibitors induce cell apoptosis by inhibiting the enzyme PARP from repairing single-strand breaks, which is the only mechanism to avoid cell death from double-strand DNA breaks, in *BRCA*-mutated patients who cannot repair DNA by homologous recombinational repair. Talazoparib, olaparib, and veliparib have been investigated for targeting in breast cancer.

The phase III EMBRACA trial (NCT01945775) explored the efficacy of talazoparib treatment in patients with *BRCA*-mutated advanced and/or MBC [85]. This study asserted the significantly higher PFS of the patients with BM. The phase III OlympiAD trial evaluated the use of olaparib for MBC with the *BRCA* mutation and demonstrated a significantly longer PFS in the olaparib group than in the standard therapy group (7.0 vs. 4.2 months; hazard ratio for disease progression or death, 0.58; 95% CI, 0.43–0.80; *p* < 0.001) [86]. 

A phase II trial (including patients with BCBM) compared cisplatin plus veliparib with cisplatin alone in BRCA-mutated BCBM (NCT02595905). The phase III BROCADE3 trial compared veliparib plus carboplatin plus paclitaxel versus placebo combined with carboplatin and paclitaxel in patients with HER2-negative advanced breast cancer and a germline *BRCA1* or *BRCA2* mutation (NCT02163694) [87]. In that study, of which 5% of patients had BCBM, the addition of veliparib to carboplatin–paclitaxel improved PFS in patients with germline BRCA mutation advanced breast cancer (hazard ratio, 0.71, 95% CI, 0.57–0.88, *p* = 0.0016).

Meanwhile, a phase I trial evaluated the safety and antitumor activity of veliparib in combination with WBRT for patients with BMs (NCT00649207) [127]. The phase II SWOG S1416 trial (NCT02595905), which explores the efficacy of veliparib plus cisplatin to treat patients with recurrent or metastatic triple-negative and/or BRCA mutation associated breast cancer with or without BMs, is ongoing.

## 4. Receptor Status/Genomic Profiling Differences between PBC and BCBM

Some studies have reported genomic sequencing data for BCBM. Recently, Morgan et al. provided a systematic review of genomic sequencing data for BCBM [128]. This review selected 13 articles on BCBM sequencing data. Twenty-two genes (*TP53*, *PIK3CA*, *KMT2C*, *RB1*, *ZFHX3*, *BRCA2*, *HER2*, *KMT2D*, *MLH1*, *PTEN*, *ATR*, *BRCA1*, *CDH1*, *COL6A3*, *FAT1*, *FLT3*, *IGFN1*, *ARID1A*, *ATM*, *CHEK2*, *MAP3K1*, and *MET*) were mutated in the BCBMs of five or more patients. Moreover, 15 (68%) of those 22 genes were actionable drug targets according to an actionability analysis.

Meanwhile, differences in receptor status/genomic profiling data between primary and BCBM have been reported in several studies. Duchnoswka et al. reported that HR conversion, particularly the loss of HR and HER2, results in changes in BCBM [129]. Schrijver et al. reported the rates of receptor discordance using a meta-analytic approach. ER conversion in the CNS was significantly higher (20.8%; 95% CI, 15.0–28.0%; *p* = 0.008) compared with liver metastasis (14.3%; 95% CI, 11.3–18.1%), but progesterone receptor conversion in the CNS (23.3%; 95% CI, 16.0–32.6%) was significantly lower than in bone metastases (42.7%, 95%CI, 35.1–50.6%, *p* < 0.001) and liver metastases (47.0%, 95%CI, 41.0–53.0%, *p* < 0.001) [130]. Hulsbergen et al. also evaluated the receptor discordance between PBC and BCBM in a large number of studies. The loss of receptor expression was associated with worse survival [131]. Interestingly, HER2 mRNA levels in BCBM were increased up to five-fold over those of PBC, and transfection of HER2 into 231-BR cells resulted in a three-fold increase in large (>50 μm^2^) BCBMs [132]. HER3 overexpression and activation of the downstream MAPK pathway were increased in BCBM compared with PBC [133]. Thomson et al. reported that 20% of patients displayed a change in ER or HER2 status in BCBM. p27kip1 and cyclin D1 and a fall in vascular endothelial growth factor A was significantly rises in BCBM [134].

The genomic profiling differences between PBC and BCBM have been analyzed in several studies. Priedigkeit et al. reported the intrinsic subtype differences between PBC and matched BCBM. Seventeen of 20 BCBMs displayed expression changes showing increases in the expressions of *FGFR4, FLT1,* and *AURKA* and the loss of *ESR1* expression [135]. Brastianos et al. reported that 53% of cases have clinically informative alterations in BCBM that are not detected in PBC [136]. Lo Nigro et al. also reported that mutations in TP53 were commonly found in CNS MBC [137]. Lee et al. reported that *TP53, PIK3CA, KIT, LH1,* and *RB1* were found in both PBC and BCBM but that the mutation frequency of *TP53* was higher in BCBM than in PBC (59.5% versus 38.9%, respectively) [138]. Tyran et al. reported similar results about the concordance of DNA copy-number alterations, mutations, and actionable genetic alterations (AGAs) between PBC and BCBM for 14 clinical pairs. Additionally, 50% of BCBM cases showed additional therapeutical AGAs not found in PBC [139]. Sato et al. reported the RNA sequencing analysis of both PBC and matched BCBM. *CXCR4*, *PLLP*, *TNFSF4*, *VCAM1*, *SLC8A2*, and *SLC7A11* were specifically up-regulated in BCBM cancer cells [140].

Those results suggest that the phenotype of BCBM is very different from that of PBC and thus clinicians must consider the appropriate treatment for BCBM. Based on those findings, a phase II clinical trial evaluating genomically guided treatment was established that investigated the use of medications targeting mutated genes in the BCBM such as abemaciclib, GDC-0084, and entrectinib (NCT03994796).

## 5. Role of Long Noncoding RNA with BCBM

The role of noncoding RNA in BCBM has also been investigated. The dysregulation of long noncoding RNAs (lncRNAs) was found to lead to the abnormal distribution of actin, thereby inducing functional changes in the BBB that were associated with cancer-derived extracellular vesicles in cell lines derived from BCBM [141]. Lnc-BM (LncRNA associated with BM) and XIST (X-inactive-specific transcript) have been reported to be associated with BCBM [142,143]. Xing et al. reported that XIST was downregulated in BCBM tissues, which led to the stimulation of the epithelial–mesenchymal transition (EMT) and promoted stemness in the tumor cells [143]. Those results suggest that this XIST-mediated pathway might be an effective targeted agent for the treatment of BCBM.

## 6. The Association between BCBM and miRNA Expression

miRNAs are small noncoding RNAs that regulate the activities of multiple genes by binding to the 3′ untranslated region of a specific gene. Extracellular vesicles contain miRNAs and may be associated with the BCBM environment [144,145]. The role of miRNAs in the metastasis processes of BCBM has been investigated in many studies. miR-7 specifically blocked BCBM in a mouse model by modulating KLF4 expression [146]. Zhang et al. reported that miR-1258 is a candidate miRNA that suppresses BCBM by targeting heparanase [147]. miR509 can modulate the genes of RhoC and TNF-α, which affect cancer cell invasion and BBB permeability, respectively; this results in the suppression of BCBM [148]. miR-20b expression was significantly associated with BCBM compared with PBC without BCBM [149]. Zhang et al. found that miR-19a in the miR-17–92 cluster from brain ascites downregulates PTEN expression in the brain environment [150].

Recently, some reports have suggested that miRNAs could play a critical role in the integrity of the BBB. miR-125a-5p has the ability to maintain the integrity of the BBB [151]. miR-181c promotes the destruction of the BBB by downregulating its target gene, *PDPK1* [152]. Bai et al. reported that miR-143 increased the permeability of human brain endothelial cells and led to the decreased expression of tight junction proteins [153].

Debeb et al. have suggested that miR-141 regulates BCBM and could be examined as a biomarker and potential target for BCBM [154]. miRNA expression has also been investigated in the cerebrospinal fluid (CSF). The levels of miR-10b and miR-21 increased significantly in the CSF of patients with BCBM. Moreover, the miR-200 family could be useful for distinguishing between types of brain cancer, such as glioblastoma and BMs [155]. 

Additionally, the role of miRNA could be associated with the epithelial mesenchymal transition (EMT), crosstalk between cancer cells and the brain microenvironment, metabolic reprogramming, and metastatic colonization in BCBM [144]. These results suggest that miRNAs have the potential to serve as therapeutic candidates for biomarkers or therapeutic targets for BCBM in the future.

## 7. The Relationship between the BRCA1/2 Mutations and BCBM

The *BRCA1* and *BRCA2* mutations are characterized by lacking an important error-free DNA repair process of homologous recombinational repair for repairing single-strand breaks; therefore, these mutations significantly increase the risks of breast cancer and ovarian cancer. Some articles have investigated the relationship between the frequency of CNS metastases and *BRCA* mutations. Lee et al. reported that there were no significant differences in CNS metastases between *BRCA1* mutation carriers and noncarriers (*p* = 0.06); however, such metastases tend to occur more frequently in those with *BRCA1* mutations (58% vs. 24%) [156]. Albiges et al. reported that patients with *BRCA1* mutations had the highest rate (67%, 10/15) of BCBM. In the same cohort, no *BRCA2* mutations were found in any of the 15 patients [157]. In a recent article, the overall rates of CNS metastasis were remarkably higher in patients with both *BRCA1* and *BRCA2* mutations than in noncarriers (*BRCA1*: 53% and *BRCA2*: 50% vs. noncarriers: 25%, respectively) [158]. Therefore, they suggest that future trials including PARP inhibitors for patients with *BRCA*-associated MBC should take the high incidence rate of CNS metastasis into account. Zavitsanos et al. evaluated the frequency rate of BCBM between groups with and without *BRCA* mutations in a matched-pair analysis [159]. They found that three-year freedom from BM was significantly shorter for group with *BRCA* mutation than group with no *BRCA* mutation (84% vs. 97%, *p* = 0.049).

## 8. Clinical Utility of Liquid Biopsy for BCBM

Liquid biopsy has been considered a potential screening tool for the earlier detection of metastasis. Determining the receptor status when screening for breast cancer metastasis is recommended according to the National Comprehensive Cancer Network (NCCN) Guidelines in Oncology for Breast Cancer [160] and the 5th ESO-ESMO (European School of Oncology-European Society for Medical Oncology) international consensus guidelines for advanced breast cancer [161]. It would be clinically important to perform a biopsy or resection based on the differences of receptor status and genomic profiling between PBC and BCBM. However, performing a biopsy or resection is difficult for some BCBM sites. Thus, a liquid biopsy might be an option for the phenotypic or genomic profiling of BCBM. Bettegowda et al. reported that ctDNA was detectable in >75% of patients with advanced breast cancer [162]. Smerage et al. investigated the effectiveness of circulating tumor cells (CTCs) for response monitoring to chemotherapy in the SWOG S0500 trial (NCT00382018) [163]. They reported the clinical significance of CTCs in patients with MBC. 

De Mattos-Arruda et al. reported that ctDNA from CNS tumors including BCBM is more abundant in the CSF than ctDNA in the plasma [164]. Siravegna et al. have reported that analyzing ctDNA from the CSF is useful for optimizing the management of HER2-positive BCBM [165]. Thus, a liquid biopsy might be an option for the phenotypic of genomic profiling of BCBM [166].

## 9. Conclusions

The incidence of brain metastasis (BM) is associated with a poor prognosis, and it remains a life-threatening condition during the course of breast cancer. However, as discussed in our review, there is no doubt that the clinical challenges in the treatment of BCBM have led to a better prognosis than ever. Treatment strategies in local therapy, including surgery and radiotherapy, are becoming less invasive and are enabling the retention of cognitive function and quality of life, which are the key clinical benefits. Meanwhile, cancer subtype-specific systemic therapies have been well-developed for breast cancer with systemic metastases other than BMs. Furthermore, novel targeted therapies have been established for BCBMs. Various clinical trials are ongoing and are expected to contribute to the better survival of patients with BCBMs in the future.

We also discussed that receptor status and genomic profiling vary between PBC and BCBM. Collecting data on these differences would be helpful in considering the potential novel therapeutic targets for patients with treatment-resistant BCBMs and would enable personalized treatment. Thus, further analysis on these differences is required more than ever. Meanwhile, various molecular mechanisms, including lncRNA, miRNA, and ctDNA have been elucidated recently. Especially, miRNAs could play potential roles in future diagnosis and treatment because they are closely associated with BCBM environment.

In our review, we focused on the latest treatment options, especially local and systemic treatments, for patients with BCBMs. Furthermore, ongoing trials and future perspectives are likely to enhance the outcomes for BCBM patients.

## Figures and Tables

**Table 1 cancers-13-01078-t001:** Major clinical trials in local treatment for BCBM.

Treatment Type	Patients’ Population	Author	Trial Name (NCT Number)	Phase	Primary Endpoint
Surgery plus WBRT vs. WBRT	Single BM	Patchell et al. [39]		III	OS
Surgery plus WBRT vs. Surgery alone	Single BM	Patchell et al. [40]		III	Recurrence of tumor in the brain
Postoperative SRS vs. WBRT	Single BM (a resected BM and a resection cavity less than 5.0cm)	Brown et al. [41]	NCCTG N107C/CEC·3 (NCT01372774)	III	Cognitive-deterioration-free survival and OS
Salvage SRS vs. postoperative WBRT	1 to 4 resected BMs with only one lesion > 3 cm	Kayama et al. [42]	JCOG0504	III	OS
WBRT alone vs. WBRT followed by SRS	1 to 3 BMs	Andrews et al. [43]	RTOG9508 (NCT00002708)	III	OS
SRS or Surgery with/without WBRT	1 to 3 BMs	Kocher et al. [44]	EORTC 22952-26001 (NCT00002899)	III	Time to PS deterioration more than 2
SRS alone vs. SRS plus WBRT	1 to 3 BMs	Brown et al. [45]	(NCT00377156)	III	Cognitive deterioration at 3 months
SRS plus WBRT vs. SRS alone	1 to 4 BMs, each under than 3 cm	Aoyama et al. [46]	(C000000412) *^,1^	III	OS
SRS for 2–4 BMs vs. 5–10 BMs	Patients with BMs who received SRS	Yamamoto et al. [47]	JLGK0901 (UMIN000001812) *^,1^	III	OS
SRS vs WBRT	4–15 untreated non-melanoma BMs	Li et al. [48]	(NCT01592968)	III	Local control rate and proportion of patients with neurocognitive decline at 4 months
SRS vs HA-WBRT	5–20 BMs	(Recruiting)	(NCT03075072)	III	Quality of life
HA-WBRT	BM outside a 5 mm margin around either hippocampus	Gondi et al. [49]	RTOG 0933 (NCT01227954)	II	Cognitive function

**Abbreviations:** BC, breast cancer; MBC, metastatic breast cancer; BM, brain metastasis; BCBM, breast cancer brain metastasis; WBRT, whole brain radiation therapy; SRS, stereotactic radiosurgery; HA-WBRT, hippocampal-avoidance whole brain radiation therapy; OS, overall survival; PFS, progression-free survival; PS, performance status. *^,1^ UMIN Clinical Trials Registry (UMIN-CTR) identifier. UMIN-CTR is an authorized clinical trial registry of the International Committee of Medical Journal Editors.

**Table 2 cancers-13-01078-t002:** Major clinical trials in systemic therapy for BCBM according to subtype.

**Systemic therapy for HER2-positive BCBM**					
**Treatment**	**Patients’ Population**	**Author**	**Trial Name** **(NCT Number)**	**Phase**	**Primary Endpoint**
Lapatinib	Progressive HER2-positive BCBM after prior trastuzumab, and cranial radiotherapy	Lin et al. [65]	EGF 105084 (NCT00263588)	II	ORR in CNS
Lapatinib plus capecitabine	HER2-positive BCBM not previously treated with WBRT, capecitabine, or lapatinib	Bachelot et al. [66]	LANDSCAPE (NCT00967031)	II	ORR in CNS
Lapatinib plus capecitabine vs. capecitabine alone	HER2-positive, locally advanced or MBC that had progressed after treatment with regimens that included an anthracycline, a taxane, and trastuzumab	Geyer et al. [67]	EGF100151 (NCT00078572)	III	Time to progression
Lapatinib plus WBRT	HER2-positive BCBM	Lin et al. [68]	(not available)	I	Maximum tolerated dose of concurrent lapatinib with WBRT
Pertuzumab plus trastuzumab and docetaxel vs trastuzumab plus docetaxel	HER2-positive locally recurrent, unresectable, or MBC without prior chemotherapy or biologic therapy for their advanced disease	Swain et al. [69]	CLEOPATRA (NCT00567190)	III	PFS
T-DM1 vs. lapatinb plus capecitabine	HER2-positive advanced breast cancer previously treated with trastuzumab and a taxane	Krop et al. [70].	EMILIA (NCT00829166)	III	Percentage of participants with progressive disease or death, PFS, OS, et al.
T-DM1	HER2-positive locally advanced or MBC with prior HER2-targeted therapy and chemotherapy	Montemurro et al. [71]	KAMILLA (NCT01702571)	III	Best overall response rate, clinical benefit rate
Neratinib plus paclitaxel vs. trastuzumab plus paclitaxel	Previously untreated recurrent and/or metastatic HER2-positive BC	Awada et al. [72]	NEfERT-T (NCT00915018)	III	PFS
Neratinib plus capecitabine	Measurable, progressive, HER2-positive BCBM	Freedman et al. [73]	TBCRC 022 (NCT01494662)	II	ORR
Neratinib plus capecitabine vs. lapatinib plus capecitabine	HER2-positive MBC with 2 or more previous HER2-directed MBC regimens.	Saura et al. [74]	NALA trial (NCT01808573)	III	PFS, OS
Afatinib alone vs. afatinib plus vinorelbine vs. investigator’s choice	HER2-positive BCBM with recurrence or progression during or after treatment with trastuzumab, lapatinib, or both	Cortés et al. [75]	LUX-Breast 3 (NCT01441596)	II	Patient benefit at 12 weeks
Tucatinib	HER2-positive MBC previously treated with trastuzumab, pertuzumab, and trastuzumab emtansine	Murthy et al. [76]	HER2CLIMB (NCT02614794)	II	PFS
Tucatinib plus T-DM1 vs. T-DM1	HER2-positive MBC previously treated with a taxane and/or trastuzumab	(Ongoing)	HER2CLIMB-02 (NCT03975647)	III	PFS
Trastuzumab Deruxtecan	HER2-positive metastatic breast cancer who had received previous treatment with trastuzumab emtansine	Jerusalem et al. [77]	DESTINY-Breast01 (NCT03248492)	II	ORR
**Systemic therapy for luminal-type BCBM**					
**Treatment**	**Patients’ Population**	**Author**	**Trial Name** **(NCT Number)**	**Phase**	**Primary Endpoint**
Taselisib plus fulvestrant vs. fulvestrant alone	ER-positive and HER2-negative locally advanced BC or MBC with recurrence or progression after aromatase inhibitor therapy	Dent et al. [78]	SANDPIPER (NCT02340221)	III	PFS
Alpelisib plus fulvestrant vs. fulvestrant alone	HR-positive, HER2-negative, advanced BC with progression after aromatase inhibitor therapy	André et al. [79]	SOLAR-1 (NCT02437318)	III	PFS
Buparlisib plus fulvestrant vs. fulvestrant alone	HR-positive, HER2-negative, locally advanced or metastatic breast cancer, who had relapsed on or after endocrine therapy and mTOR inhibitors	Di Leo et al. [80]	BELLE-3 (NCT01633060)	III	PFS
Palbociclib	Measurable progressive luminal-type BCBM	(Ongoing)	(NCT02896335)	II	Clinical benefit rate at 8 weeks
Abemaciclib	BM from luminal-type BC, NSCLC, or melanoma	(Ongoing)	(NCT02308020)	II	ORR in CNS
Palbociclib plus trastuzumab plus lapatinib plus fulvestrant	ER-positive/HER2-positive BCBM	(Ongoing)	(NCT04334330)		ORR
Abemaciblib plus SRS vs. palbociclib plus SRS vs. ribociclib plus SRS	ER-positive/HER-2 negative BCBM	(Ongoing)	(NCT04585724)	I	Incidence of grade 3+ radiation therapy oncology central nervous system toxicity
**Systemic therapy for TN-type BCBM**					
**Treatment**	**Patients’ Population**	**Author**	**Trial Name** **(NCT Number)**	**Phase**	**Primary Endpoint**
Pembrolizumab	Advance TNBC, advanced head and neck cancer, advanced urothelial cancer, or advanced gastric cancer	Nanda et al. [81]	KEYNOTE-012 (NCT01848834)	I	Adverse events and overall response rate
Atezolizumab plus nab-paclitaxel vs. nab-paclitaxel	unresectable locally advanced or metastatic TNBC	Schmid et al. [82]	IMpassion130 (NCT02425891)	III	PFS and OS
Carboplatin and bevacizumab	New or progressive BCBM	Leone et al. [83]	(NCT01004172)	II	ORR in CNS
Bevacizumab, etoposide, cisplatin	Breast cancer brain and/or leptomeningeal metastasis	Wu et al. [84]	(NCT01281696)	II	ORR in CNS
Talazoparib vs. single agent chemotherapy investigator’s choice	Advanced and/or MBC patients with BRCA mutation, which received no more than 3 prior chemotherapy-inclusive regimens for locally advanced and/or metastatic disease	Litton et al. [85]	EMBRACA (NCT01945775)	III	PFS
Olaparib vs. single agent chemotherapy investigator’s choice	MBC who had received no more than two previous chemotherapy regimens for metastatic disease	Robson et al. [86]	OlympiAD (NCT02000622)	III	PFS
Veliparib plus carboplatin plus paclitaxel vs. carboplatin plus paclitaxel	Advanced HER2-negative breast cancer with BRCA1 or BRCA2 mutation	Diéras et al. [87]	BROCADE3 (NCT02163694)	III	PFS

**Abbreviations:** BC, breast cancer; MBC, metastatic breast cancer; BM, brain metastasis; BCBM, breast cancer brain metastasis; WBRT, whole brain radiation therapy; SRS, stereotactic radiosurgery; HER2, human epidermal growth factor receptor 2; TNBC, triple-negative breast cancer; OS, overall survival; PFS, Progression-Free Survival; ORR, objective response rate; PS, performance status; CNS, central nervous system; T-DM1, trastuzumab emtansine; NSCLC, non-small cell lung cancer; BRCA, breast cancer gene.

## Data Availability

The data presented in this study are available in Appendix A.

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
