# Peer review of "Breast Cancer Brain Metastasis—Overview of Disease State, Treatment Options and Future Perspectives"

_cancers, 2021, doi:10.3390/cancers13051078_

Round 1

Reviewer 1 Report

In this manuscript, the authors provide a very comprehensive review of current treatments aiming at diagnosing and treating breast cancer brain metastasis with a focus on recent clinical trials. The review is generally well written and references are adequate.

Major comments:

  1. There is an inconsistency throughout the manuscript related to the amount of details used to describe studies and results. Some sections are very informative with proper explanation of the study settings and discussion of outcomes while other sections appear more as a list of references with no content. I would recommend to restrain the focus of the review and elaborate mostly on local and systemic treatments (sections 6 and 7) which are already quite extensive. Previous sections (1-5) could be fused in one large introduction section (with subtitles if you wish) in which the authors could engage in a brief description of incidence, risk factors, prognosis and BBB crossing. The introduction should be more of a discussion of the topic rather than a listing of all studies with little to no details.
  2. The conclusion section is very brief and does not offer a real perspective from the authors. I would recommend improving this section by including the discussions on new treatments such as genomic profiling, long non-coding RNA and miRNA in this section. The authors should also provide their opinion and/or vision on what they think represent the most promising avenue for the treatment of this disease and where the field should go from now.

Minor comments:

  1. Why are there no references at all in section 1 introduction?
  2. Line 70: subtype of what? Cancer subtype would read better.
  3. Line 129: there is a typo in the name of the author ‘’Pardridge’’.
  4. Table 2: The first two treatments have the same primary endpoint but it is worded differently in the table. Both should read ‘’CNS objective response’’ for consistency.
  5. Line 329: There is a typo ‘’pus’’ should be ‘’plus’’.
  6. Line 404: Provide more details about the results. Delayed by how much? Was it significant?

Author Response

Response to Reviewer 1:

Dear Reviewer 1:

We thank you for your insightful comments regarding our manuscript, which have helped us improve our report.

Major comments:

Comment1:

There is an inconsistency throughout the manuscript related to the amount of details used to describe studies and results. Some sections are very informative with proper explanation of the study settings and discussion of outcomes while other sections appear more as a list of references with no content. I would recommend to restrain the focus of the review and elaborate mostly on local and systemic treatments (sections 6 and 7) which are already quite extensive. Previous sections (1-5) could be fused in one large introduction section (with subtitles if you wish) in which the authors could engage in a brief description of incidence, risk factors, prognosis and BBB crossing. The introduction should be more of a discussion of the topic rather than a listing of all studies with little to no details.

Response:

Thank you for your recommendation.

We have revised the Introduction section as per your suggestion in the major comment. We have increased the length of the Introduction section by adding more references to the former Incidence rate section, which has been changed to BBB section, resulting in a shorter introduction before the discussion on the treatment. This will contribute to more focus on the treatment sections.

To be more specific, we have omitted almost all of the Introduction section’s original content and have shortened the insistent description of HER2 type in the Incidence rate section (Line 55). We have summarized the risk factors more briefly (Line 63) and reduced the descriptions on prognostic factors (Line 80).

Comment2:

The conclusion section is very brief and does not offer a real perspective from the authors. I would recommend improving this section by including the discussions on new treatments such as genomic profiling, long non-coding RNA and miRNA in this section. The authors should also provide their opinion and/or vision on what they think represent the most promising avenue for the treatment of this disease and where the field should go from now.

Response:

Thank you for your recommendation.

We have revised the Conclusion section by adding more information to the previous sections (Lines 657–679). In the first half, we have reviewed the current state of BCBM and the previous and ongoing clinical trials of local and systemic treatment. In the latter half, we have reviewed mainly the basic research on receptor status, genomic profiling, and early detection of BCBM with recently discovered molecular mechanisms. Lastly, we have referred to the prospects for future studies.

Minor comments:

Comment 1:

Why are there no references at all in section 1 introduction?

Response:

Thank you for comment. As previously described in our response to the major comment, we have revised the Introduction section with additional references (Lines 38–106).

Comment 2:

Line 70: subtype of what? Cancer subtype would read better.

Response:

Thank you for your comment. We have added “cancer” before “subtype” (Line 58).

Comment 3:

Line 129: there is a typo in the name of the author ‘’Pardridge’’.

Response:

Thank you for bringing this to our attention.

We have corrected the author’s name to “Pardridge” (Line 86).

Comment 4:

Table 2: The first two treatments have the same primary endpoint but it is worded differently in the table. Both should read ‘’CNS objective response’’ for consistency.

Response:

Thank you for your suggestion.

We have ensured consistency in the terms used, “ORR in CNS,” for the same primary endpoint (Table 2). We have described ORR as an abbreviation for objective response rate.

Comment 5:

Line 329: There is a typo ‘’pus’’ should be ‘’plus’’.

Response:

Thank you for bringing this to our attention.

We have corrected “pus” to “plus” (Line 300).

Comment 6:

Line 404: Provide more details about the results. Delayed by how much? Was it significant?

Response:

Thank you for your suggestion.

We have added descriptions of the incidence rates of BCBM decline and delay (Lines 375 and 376).

Other revision points

We apologize that we have revised this manuscript with multiple points as follows in order to get better understanding of each article.

We have also added the values of the statistical parameters (e.g., hazard ratio, 95% CI, and p value) for the Local treatment and Systemic therapy sections. We have also replaced “HR” with “hazard ratio” to distinguish it from “hormone receptor” (Lines 128, 135, 138, 142, 151, 153, 154, 157, 163, 167, 168, 171, 173, 179, 182, 196, 200, 209, 210, 210, 211, 212, 221, 227, 235, 297, 298, 311, 320, 330, 331, 349, 367, 373, 374, 396, 422, 427, 429, 473, 520, and 523).

We have added UMIN-CTR and correct descriptions about each trial for better understanding (Table 1, Tabe2, Table S2).

We have added EMILIA, DESTINY-Breast 01, BELLE-3 trials, and study name EGF100151 (Table2).

We have rearranged TableS2 for better understandings.

We have shortened the description on trastuzumab to avoid duplications of the previous descriptions (Line 250).

We have revised the results of the HER2CLIMB trial (tucatinib) to rectify the statistical data (Line 398).

We have omitted the BCBM data of the DESTINY-Breast01 trial because the description was presented only at the ESMO meeting (Line 405).

We have added further information about TILs (Lines 473, 475).

We have corrected the PD-L1 expression rate (Line 476).

We have omitted KEYNOTE-0208 study for consistency (Line 480).

We have revised PD-L1 to immune checkpoint inhibitors (Line 487).

We have omitted the description about cancer subtype of reference [83] for better understanding of this paragraph (Line 500).

We have corrected “with BM” to “without BM” in accordance with the EMBRACA trial data (Line 512).

We have omitted the BCBM data of the OlympiAD trial because the description was presented only at the ASCO meeting (Line 521).

We have corrected author name from Nigro to Lo Nigro (Line 561)

We have rearranged miR-125a-5p for consistency (Line 602).

Reviewer 2 Report

This is an extensive review on breast cancer brain metastasis.

I suggest only to add 2 recent references as follows:

  • In the paragraph 5 (The role of BBB...) to replace the reference n.31 with the more recent one which is specific fro brain metastases :  Soffietti R, Ahluwalia M, Lin N, Rudà R. Management of brain metastases according to molecular subtypes. Nat Rev Neurol. 2020 Oct;16(10):557-574. doi:10.1038/s41582-020-0391-x. Epub 2020 Sep 1. PMID: 32873927.
  • In the paragraph 12 (Clinical utility of liquid biopsy..) to add at the end of the statement "Thus, a liquid biopsy might be an option for the phenotypic of genomic profiling of BCBM" the recent RANO review :  Boire A, Brandsma D, Brastianos PK, Le Rhun E, Ahluwalia M, Junck L, Glantz M, Groves MD, Lee EQ, Lin N, Raizer J, Rudà R, Weller M, Van den Bent MJ, Vogelbaum MA, Chang S, Wen PY, Soffietti R. Liquid biopsy in central nervous system metastases: a RANO review and proposals for clinical applications. Neuro Oncol. 2019 May 6;21(5):571-584. doi: 10.1093/neuonc/noz012. PMID: 30668804; PMCID: PMC6502489.

Author Response

Response to Reviewer 2:

Dear Reviewer 2:

We thank you for your insightful comments regarding our manuscript, which have helped us improve our report.

Reviewer 2

I suggest only to add 2 recent references as follows:

  • In the paragraph 5 (The role of BBB…) to replace the reference n.31 with the more recent one which is specific fro brain metastases:  Soffietti R, Ahluwalia M, Lin N, Rudà R. Management of brain metastases according to molecular subtypes. Nat Rev Neurol. 2020 Oct;16(10):557-574. doi:10.1038/s41582-020-0391-x. Epub 2020 Sep 1. PMID: 32873927.
  • In the paragraph 12 (Clinical utility of liquid biopsy.) to add at the end of the statement "Thus, a liquid biopsy might be an option for the phenotypic of genomic profiling of BCBM" the recent RANO review:  Boire A, Brandsma D, Brastianos PK, Le Rhun E, Ahluwalia M, Junck L, Glantz M, Groves MD, Lee EQ, Lin N, Raizer J, Rudà R, Weller M, Van den Bent MJ, Vogelbaum MA, Chang S, Wen PY, Soffietti R. Liquid biopsy in central nervous system metastases: a RANO review and proposals for clinical applications. Neuro Oncol. 2019 May 6;21(5):571-584. doi: 10.1093/neuonc/noz012. PMID: 30668804; PMCID: PMC6502489.

Response:

Thank you for your recommendations.

We have replaced the original references with the references suggested by you (Line 82 [25] and Line 655 [163]).

Other revision points

We apologize that we have revised this manuscript with multiple points as follows in order to get better understanding of each article.

We have also added the values of the statistical parameters (e.g., hazard ratio, 95% CI, and p value) for the Local treatment and Systemic therapy sections. We have also replaced “HR” with “hazard ratio” to distinguish it from “hormone receptor” (Lines 128, 135, 138, 142, 151, 153, 154, 157, 163, 167, 168, 171, 173, 179, 182, 196, 200, 209, 210, 210, 211, 212, 221, 227, 235, 297, 298, 311, 320, 330, 331, 349, 367, 373, 374, 396, 422, 427, 429, 473, 520, and 523).

We have added UMIN-CTR and correct descriptions about each trial for better understanding (Table 1, Tabe2, Table S2).

We have added EMILIA, DESTINY-Breast 01, BELLE-3 trials, and study name EGF100151 (Table2).

We have rearranged TableS2 for better understandings.

We have shortened the description on trastuzumab to avoid duplications of the previous descriptions (Line 250).

We have revised the results of the HER2CLIMB trial (tucatinib) to rectify the statistical data (Line 398).

We have omitted the BCBM data of the DESTINY-Breast01 trial because the description was presented only at the ESMO meeting (Line 405).

We have added further information about TILs (Lines 473, 475).

We have corrected the PD-L1 expression rate (Line 476).

We have omitted KEYNOTE-0208 study for consistency (Line 480).

We have revised PD-L1 to immune checkpoint inhibitors (Line 487).

We have omitted the description about cancer subtype of reference [83] for better understanding of this paragraph (Line 500).

We have corrected “with BM” to “without BM” in accordance with the EMBRACA trial data (Line 512).

We have omitted the BCBM data of the OlympiAD trial because the description was presented only at the ASCO meeting (Line 521).

We have corrected author name from Nigro to Lo Nigro (Line 561)

We have rearranged miR-125a-5p for consistency (Line 602).